# A Comparison of Ocean Model Results with Satellite Observations during the Development of the strong 1997-98 El Niño

**David J. Webb, Andrew C. Coward, and Helen M. Snaith**

National Oceanography Centre, Southampton SO14 3ZH, U.K.

**Correspondence:** D.J.Webb (djw@noc.ac.uk)

**Abstract.**

Descriptions of the ocean's role in the El Niño usually focus on equatorial Kelvin waves and the ability of such waves to change the mean thermocline depth and the sea surface temperature (SST) in the central and western Pacific.

In contrast, starting from a study of the transport of water with temperatures greater than 28°C, sufficient to trigger deep atmospheric convection, Webb (2018) found that during the strong El Niños of 1983-1984 and 1997-1998, advection by the North Equatorial Counter Current (NECC) had a much greater impact on sea surface temperatures than processes occurring near the Equator.

Webb's analysis, which supports the scheme proposed by Wyrtki (1973, 1974), made use of archived data from a high resolution ocean model. Previously the model had been checked in a preliminary comparison against SST observations in the equatorial Pacific, but given the contentious nature of the new analysis, the model's behaviour in key areas need to be checked further against observations.

In this paper this is done for the 1987-1988 El Niño, making use of satellite observations of SST and sea level. SST is used to check the movement of warm water near the Equator and at the latitudes of the NECC. Sea level is used to check the model results at the Equator and at 6°N in the North Equatorial Trough. Sea level differences between these latitudes affect the transport of the NECC, the increase transport at the start of each strong El Niño being associated with a drop in sea level at 6°N in the western Pacific. Later rises in sea level at the Equator increases the transport of the NECC in mid-ocean.

The variability of sea level at 6°N is also used to compare the strength of tropical instability waves (TIWs) in the model and in the observations. The model showed that in a normal year these act to dilute the temperature in the core of the NECC. However their strength declined during the development of the strong El Niños, allowing the NECC to carry warm water much further than normal across the Pacific.

---

## 1  Introduction

Wyrtki (1973, 1974) analysed sea level time series at Pacific islands and showed that there was a correlation between the occurrence of strong El Niños and an increase in the sea level difference across the North Equatorial Counter Current. This current lies near 5°N and carries near surface ocean water from the warm western equatorial Pacific into the eastern Pacific, north of the Galapagos.

The increased sea level difference implies an increased pressure difference within the ocean and, as with pressure differences in the atmosphere, this in turn implies an increased transport by the current. Wyrtki surmised that the increased current would transport more warm surface water from the western to the eastern Pacific. Here it would trigger strong convection in the atmosphere and so trigger an El Niño.

Unfortunately the work was published at a time when analytic and numerical models of the equatorial ocean were leading researchers in another direction. Thus Anderson and Rowlands (1976) and McCreary (1978) both showed that Kelvin waves caused upwelling on Central America, the implication being that this could cause warming of the surface layer sufficient to generate an El Niño (Gill, 1982).

This led to an emphasis on the potential role of equatorial Kelvin waves in triggering El Niños. Indeed Wyrtki's own theory of the El Niño (Wyrtki, 1975) states that "the accumulated water flows eastward, probably in the form of an internal equatorial Kelvin wave" and "This wave leads to an accumulation of warm water off Ecuador and Peru".

Other papers which use the Kelvin wave hypothesis include McPhaden (1999), Vialard et al. (2001) and more recently Levine and McPhaden (2016), Chen et al. (2016) and Hu and Fedorov (2017).

However although equatorial Kelvin waves can readily transport energy and momentum, wave motions are very inefficient at transporting mass and related properties such as salinity and heat. Kelvin waves might trigger another process which heat the central and eastern Pacific but although internal Kelvin waves can warm the deep ocean through downward advection, there is no way in which the reverse process can warm the surface layer.

This problem of the heat transport during an El Niño, was investigated by Webb (2018) using archived data from a 1/12°global ocean model. The study concentrated on the strong El Niños of 1982-83 and 1997-97 and found that equatorial Kelvin waves had no significant effect on the surface temperature of the eastern Pacific.

Instead the model results showed that both the temperature and volume of water carried by the North Equatorial Counter Current (NECC) increased during the period that the El Niños were growing.

This resulted in water reaching the eastern Pacific with temperatures above 28°C, sufficient to trigger deep atmospheric convection (Evans and Webster, 2014). This occurred near the latitude of the Inter Tropical Convergence Zone (ITCZ), the band where the where the atmosphere is more unstable than normal. As a result it was concluded that it was this water which triggered the strong El Niños.

## 1.1   Mechanisms

The model results also indicated that three physical mechanisms are involved. The first is an increased strength of the NECC which is in part due to the annual Rossby wave (Myers, 1979). This starts in the eastern Pacific near the beginning of each year and reaches the dateline about five months later. In mid-1982 and mid-1997 it appears to progress much further westwards than normal but this may be a local west Pacific effect increasing the depth of the North Equatorial Trough. Whatever the cause, the changes in the depth of the trough increase the pressure difference across the NECC and at some longitudes move it towards the Equator, both processes increasing its transport.

It is the increased transport in the west equatorial Pacific which appears to start the strong El Niños of 1982-83 and 1997-98. The NECC always carries some warm pool water to the east, but this extra impetus means that it carries warm water much further eastwards then normal. As the water is above 28°C it encourages new centres of deep atmospheric convection to develop over the ocean. The result supports Wyrtki's observations (Wyrtki, 1973, 1974) discussed above.

The second mechanism involves an increase in the core temperature of the NECC due to the reduction in the strength of tropical instability waves (or eddies). In a normal year the eddies extract warm water from the core of the NECC and replace it with cooler water, so reducing the temperature of the current.

However as the El Niños develop and the areas of deep atmospheric convection move eastwards, the easterly winds along the equator decay to zero and, as convection moves even further east, they are replaced by westerlies. The surface current along the Equator reflects this, the westward flowing Equatorial current decaying to zero and eventually being replaced by a eastward flowing current.

A key result of this is that the Equatorial Current can no longer power the tropical instability eddies. These die away and no longer dilute the NECC. As a result the NECC carries warm water further east, again triggering a new region of deep atmospheric convection and so repeating the process.

Finally the model confirmed a third mechanism, originally proposed by Kug et al. (2009), where the region of highest sea level on the Equator, moves from the western Pacific into the central Pacific. Comparison with the wind field indicates that this develops between the westerly and easterly winds along the Equator. The increased sea level on the Equator in this region also increases the pressure difference across the NECC and so further increases its transport of warm surface water towards the eastern Pacific.

## 1.2   Comparison with Observations

But these are just model results and, given the widespread acceptance of the Kelvin wave ideas discussed earlier[1], both the results and the mechanisms proposed need to be carefully checked against observations.

To a certain extent this has been done. Webb (2016) investigated surface temperatures in the Pacific Niño regions and found good agreement between the model and observations. That study also investigated whether the changes occurring during an El Niño were due to advection, other model processes or feedback between the model and the atmospheric forcing. The results indicated that any feedback was cooling the ocean surface and that surface warming during the development of the strong El Niños was due primarily to advection.

However this study was carried out before the results of Webb (2018) were known, and given the contentious nature of the latter's results, the model's behaviour in key areas need to be checked further against observations. If serious discrepancies are found then the conclusions of the paper become invalid.

In the present study this is done by using satellite data from 1995-2000 to check the model sea surface temperature (SST) and sea surface height (SSH) fields in regions most affected by the three mechanisms discussed above.

---

[1]See also Wikipedia "Equatorial wave" (version of 21 September 2019).

When comparing the SST fields, the present paper uses absolute values because, as discussed by Evans and Webster (2014), it is only when the SST rises above 28°C, that deep atmospheric convection can occur. SSH is useful because, in the model, the annual Rossby wave and tropical instability waves show up clearly in the sea surface height signal. As a result, satellite based radar altimeters, which measure sea level to within a few centimetres, provide a useful check on both the Rossby wave and the tropical instability waves seen in the model.

There were no altimeter measurements of SSH during the 1982-83 period but by 1997-98, both the Topex-Poseidon and ERS-2 satellite altimeters were operating. This paper thus concentrates on the later period.

In the remainder of this paper section 2 gives details of the model and satellite data, and the processing used to generate the gridded data sets used later. Section 3 then focuses on the sea surface temperature signal in the equatorial band (used in many El Niño analyses) and also in a band including the core of the NECC. Comparisons between model and observations are also made for September 1997 when the El Niño is developing rapidly and for September 1996, during a more typical year.

Section 4 then compares the observed and model sea levels at 6°N, a key latitude for the propagation of the annual Rossby wave, and at the Equator. At 6°N the tropical instability eddies show up as propagating waves so this property is used in Section 5 to check that the strength of the eddies in the model is comparable to those in the real world.

Overall qualitative agreement between model and observations is good for both SST and sea level. However in the case of the instability eddies, when these are weakest, the variance in the model data is smaller than in the observations - as though the model is too quiet.

The final section reviews the results and discusses how they support, or fail to support, the proposed mechanisms controlling the growth of a strong El Niño.

## 2  Data Sources and Processing

The model data used in this paper is from the five day average datasets generated during run 6 of the Nemo 1/12°model. This is the same data as used for Webb (2016) and Webb (2018), where detailed information on the model, the model run and the archive datasets can be found. The model data is on a 1/12 degree grid but, for the comparisons reported here, sea surface values were averaged onto the same grid as used for the processed satellite data.

The processed sea surface temperature used in this paper is the weekly averaged version of the Reynolds Optimally Interpolated satellite data (Reynolds and Stokes, 1981). The analysis scheme, described in Reynolds et al. (2002), combines data from separate satellites and corrects for the effects of clouds and other errors in the measurements. The data is provided in the form of averages over one-degree cells.

The processed satellite altimeter data is the Copernicus DUACS DT2014 gridded absolute dynamic topography dataset (Pujol et al., 2016; Taburet and Team, 2018, 2018). For the period studied here this combines the altimeter measurements of the ERS-2 and Topex/Poseidon satellites, correcting for errors and using optimal interpolation to generate data on a 1/4-degree grid at daily intervals.

There are some reservations about the altimeter data which need to be kept in mind. Pujol et al. (2016) refer to error variances of up to 32.5 cm$^2$ in energetic regions of ocean and Taburet and Team (2018) (2018) state that two satellites are the "minimum for offline applications". They also show that the effective spatial resolution of the gridded data may be 200 km and more in the equatorial regions discussed here.

## 3  Comparison of Sea Surface Temperatures

Figure 1 compares the sea surface temperature (SST) averaged between 5°S and 5°N for the period 1995 to 2000. Both the model and observations show the 1997-98 El Niño developing in two stages, an initial advance of warm water towards the central Equatorial Pacific between March and June 1997, followed by the main El Niño advance which starts during late (northern) summer and reaches the eastern boundary of the Pacific in early 1998.

As discussed in Webb (2018) the first advance may have been triggered by winds crossing the Equator north of New Guinea. The following main advance is then consistent with the development of an enhanced North Equatorial Counter Current (NECC) arising from a stronger than normal pressure difference across the current.

The results show that the observations and model are generally in good agreement in both the magnitude and the timing, not only of the El Niño but also of the many other features seen in the figures. For the El Niño, the main discrepancy occurs in the central Pacific where the difference figure shows that the model is up to one degree warmer than the observations.

The model is also warmer than observations in the eastern Pacific where there appears to be a jump in model temperatures east of the Galapagos Islands at 270°E. This may be due to the model topography, the relatively coarse model representation of the islands partly blocking both the Equatorial Current and Equatorial Undercurrent.

In addition the figures show that, outside the El Niño years, the central Pacific temperature anomalies due to tropical instability waves appear larger in the model data set than in the observations. However after allowing for this, there is agreement with the model in both the east-west wavelength of the waves and the variations in the speed as they propagate westwards. The differences in the amplitudes may be due to

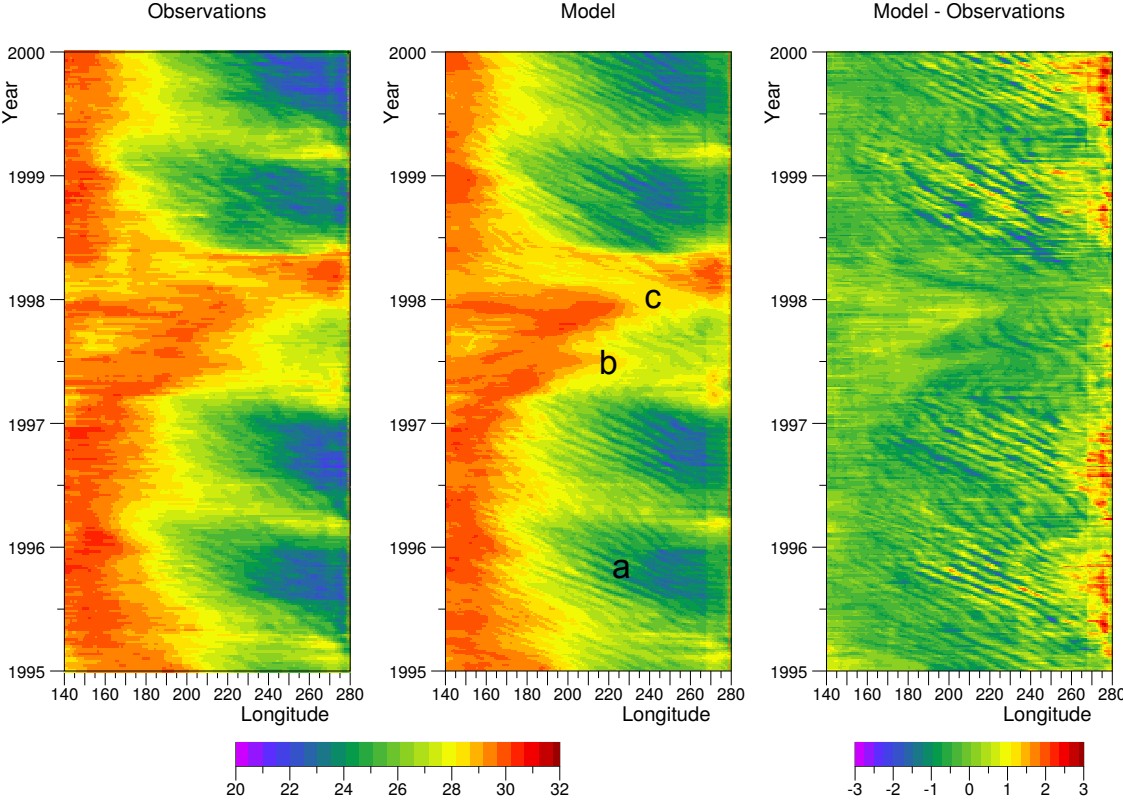

**Figure 1.** Observed and model sea surface temperatures and their difference, averaged between 5°S and 5°N, for the equatorial Pacific between 140°E and 280°E (80°W). Showing (a) Tropical Instability Waves, strongest in second half of non-El Niño years, (b) initial mid-ocean El Niño, discussed in Webb (2018), (c) strong El Niño, starting in the west around mid-year and arriving in the east around new year. Temperature units are degrees C.

model errors but they could also be due to smoothing of the observations by the optimal interpolation process.

The waves are emphasized in the difference plot, as would be expected for chaotic events. In this respect the lack of any short wavelength features in the difference plot during the second half of 1997, when the El Niño was growing, is significant.

The mechanisms proposed by Webb (2018) mainly involved the transport of heat by the NECC. Unfortunately the current lies partly outside the equatorial band of Fig 1, so as a check Fig. 2 shows a similar comparison for the latitude range 3°N and 9°N. This includes most of the NECC and is centred on the 6°N latitude discussed later. It also covers most of the latitude range of the ITCZ where the atmosphere appears to be most unstable.

The figures again illustrate good agreement between model and observations and, as discussed below, show that during the development of the El Niño, when the central Pacific warms, there is a cooling of the western Pacific at these latitudes.

The tropical instability waves between 3°N an 9°N are slightly weaker than those of Fig. 1, but in both cases the difference plot shows that during the El Niño they all but disappear in the central Pacific.

## 3.1  Surface Temperature Maps

The El Niño is usually characterised in terms of the average sea surface temperatures between 5°N and 5°S. However the model results reported by Webb (2018) indicate that the most significant changes occur further north.

Here, during the development of an El Niño, the North Equatorial Counter Current can transport significant amounts of warm water into the eastern Pacific near the latitude of the atmospheric Inter-Tropical Convergence Zone (ITCZ). There the warm water can trigger enhanced deep atmospheric convection and thus be the cause of a strong atmospheric El Niño.

Figure 3 compares the observed and model fields of SST during late September in 1996, a normal year, and in 1997, during the development of the 1997-98 El Niño.

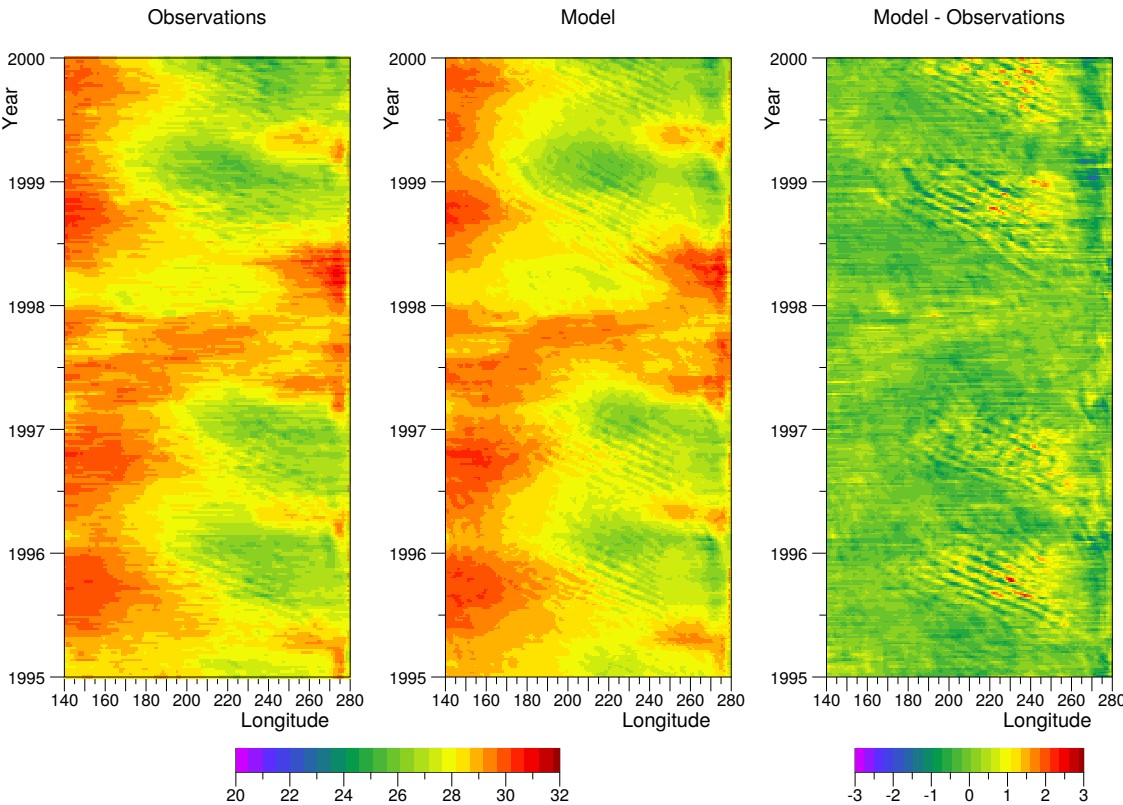

**Figure 2.** Observed and model sea surface temperatures and their difference, averaged between 3°N and 9°N, for the equatorial Pacific between 140°E and 280°E (80°W). Units are degrees C.

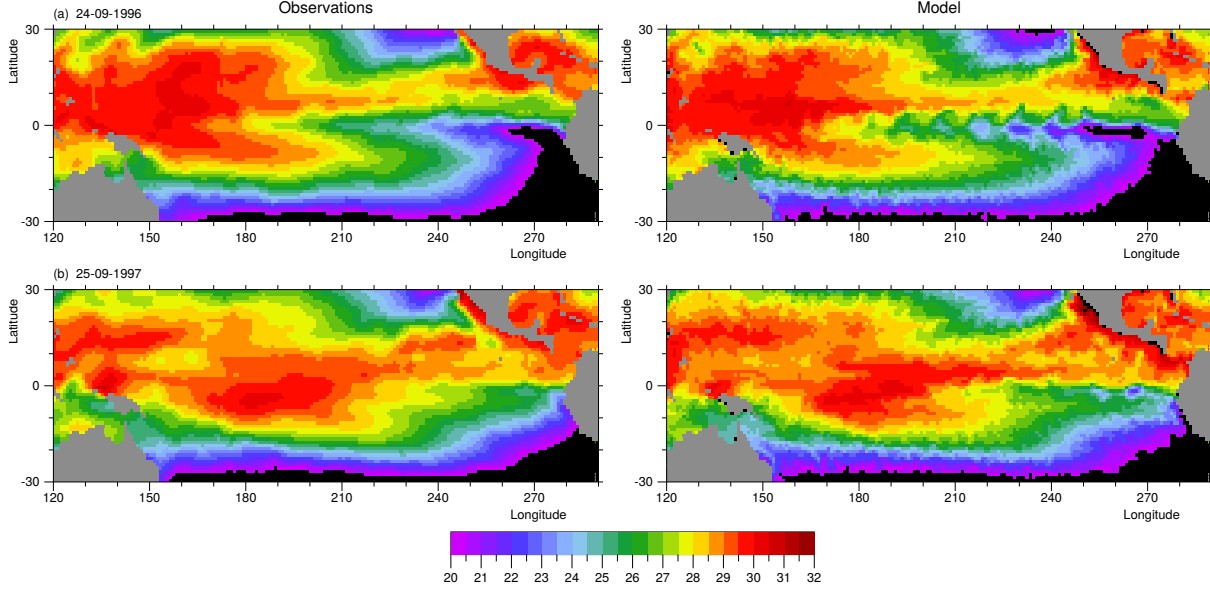

**Figure 3.** Observed and Model Sea Surface Temperatures in (a) late September 1996 and (b) late September 1997. Units are degrees C. For these figures the model data has been averaged onto the same one-degree grid as used for the satellite based dataset.

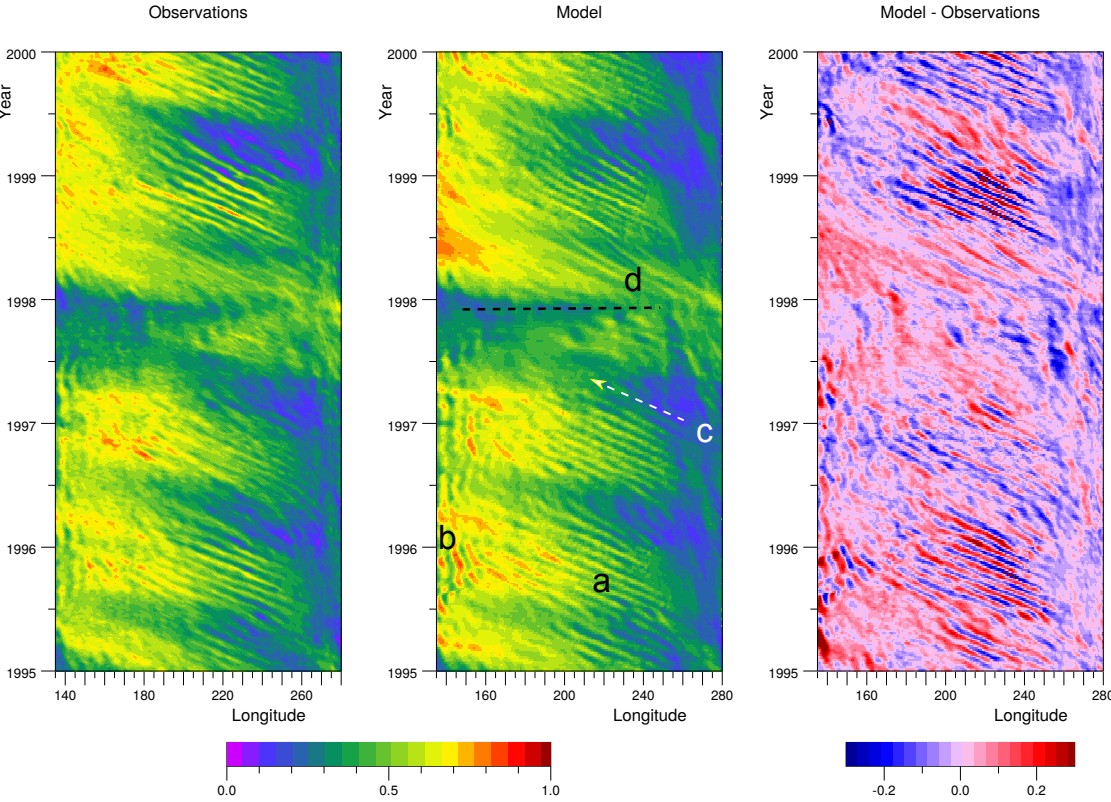

**Figure 4.** Observed and model sea surface height and their difference in the Pacific at latitude of 6°N between 135°E and 280°E (80°W). Showing (a) westward propagating tropical instability waves or eddies, (b) quasi-steady meanders near the start of the NECC, (c) westward propagating annual Rossby wave, (d) Transition Event (see text). Units are metres.

Late September is of interest because, as discussed in Webb (2018), during the development of a strong El Niño it shows two important features.

The first, the result of enhanced transport by the NECC is the region of enhanced SST north of the Equator in the eastern Pacific. The second is the region of high SSTs that develops in the central Pacific. As is discussed later, the latter also shows a maximum in sea level, probably because it lies between regions of easterly and westerly winds acting along the Equator.

The results for September 1996, show generally good agreement. The most obvious difference occurs along the Equator in the central and eastern Pacific where the the Tropical Instability waves are much more pronounced in the model plot than in the observations. Plots of individual satellite SST measurements, for example Wentz et al. (2000), usually show much stronger instability waves, so it is possible that the Reynolds SST processing has smoothed out these features.

There is also good agreement in September 1997. Both the observations and the model show a region of high temperatures at the latitude of the NECC which extends from the central Pacific into the eastern Pacific. The model results

showed that this water had been advected from the west. In support of this, both figures show a region of reduced SSTs between 150°E and 170°E at the latitudes of the NECC.

Both observations and model also show the eastward shift in the region of highest SSTs along the Equator. This shift also affects temperatures south of the Equator as far as 10°S.

## 4   Sea Levels along 6°N and the Equator

Webb (2018) found that the increased strength of the NECC during the growth of an El Nino was in part associated with an increased sea level difference between the Equator and 6°N. It is thus of interest to see how the observations and model compare at both these latitudes.

Figure 4 compares the observed and model sea levels at 6°N (i.e. the gridded data averaged between 5.75°N and 6.25°N). Between 1995 and 1997 both behave in a similar manner, showing a similar mean east-west slope, an annual signal and small scale structures. Between 160°E and 250°E these are due to westward propagating tropical instability

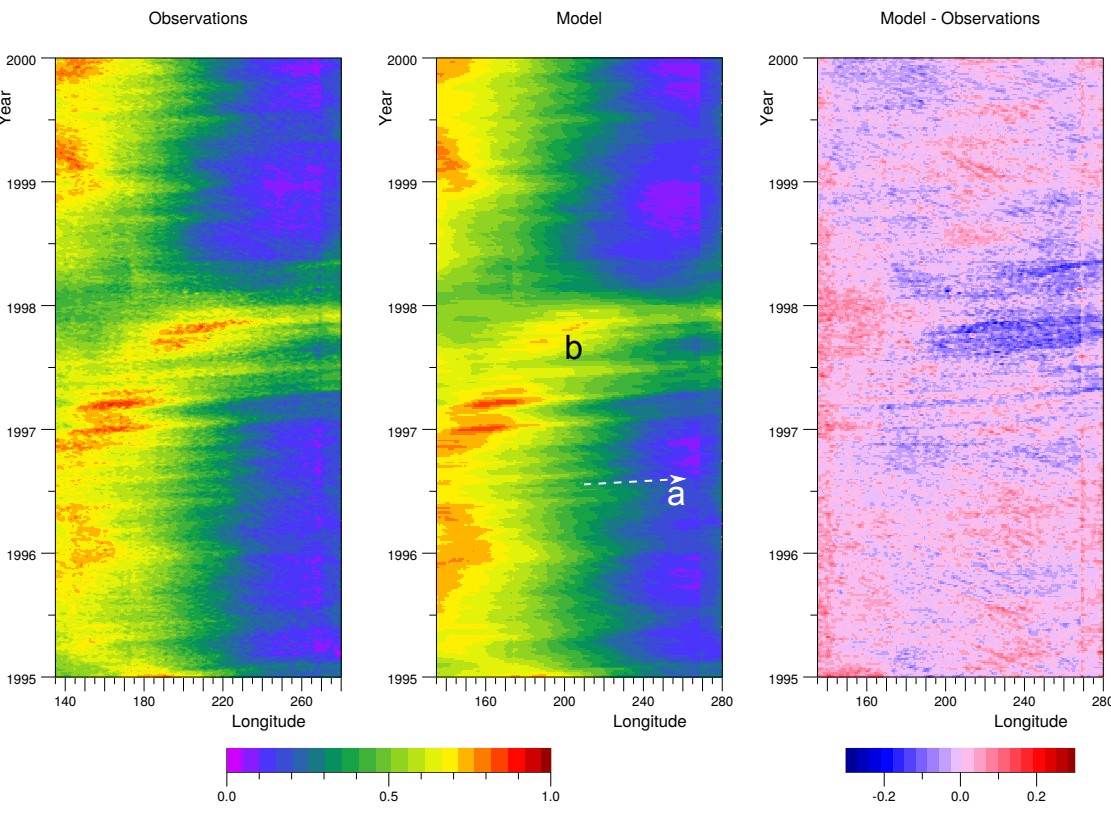

**Figure 5.** Observed and model sea surface height and their difference in the Pacific on the Equator between 135°E and 280°E (80°W). Showing (a) eastward propagating Kelvin waves, (b) sea level high moving to the central Pacific. Units are metres.

waves. West of 160°E they are due to the, almost steady, north-south meanders generated near the start of the NECC.

The years 1995 and 1996 also show the annual Rossby wave propagating westwards in both observations and model. This also occurs in 1997 but in the second half of the year both the observations and model show and additional drop in sea level that between 130°E and 180°E. Webb (2018) argues that it was this drop in sea level which was responsible increasing the transport of the NECC, allowing it to carry warm surface water further east than normal and so trigger the strong El Niño..

Sea levels in the west continue dropping towards the end of 1997, but then there is a transition which sea level starts rising from the western boundary as far as 250°E. This is also the period when the raised sea level along the Equator starts returning to more normal values (Fig. 6) and there are other major changes in both the Equatorial Current and the NECC (Webb, 2018). The transition is most probably the results of the southward movement of the ITCZ (see: Wodzicki and Rapp, 2015), the trade winds of the North Pacific reaching

and sometimes crossing the Equator during the early months of 1998[2].

During 1998 the standing waves due to the NECC are weaker than in the previous years as initially are the tropical instability waves. The latter reappear strongly during the second half of the year after which the ocean appears to return to its normal state.

The comparison between observations of SSH and the model results at the Equator is shown in Fig. 5. The agreement is good, both in the large scale structure and the timing of individual Kelvin waves seen to propagate eastwards across the Equatorial Ocean. Although the timing of these events seems very good, they will be a direct response to the wind forcing and so the agreement is really a measure of the quality of the wind forcing used to drive the model.

Possibly of more importance is the close agreement in the strength and position of the region of maximum sea level which, during 1997, moves from the western boundary into the centre of the Pacific. During the second and third quarters of 1997 this will have increased the sea level difference in the

[2]A similar transition is seen at the end of 1983 during the strong 1983-1984 El Niño.

western and central Pacific between the equator and 6°N. As a result it will have been partly responsible for the enhanced strength of the NECC.

## 5 Mixing by Tropical Instability Waves

Webb (2018) found from the model results that the transport of warm water by the NECC during an El Niño was also increased as a result of reduced mixing by tropical instability eddies. In the paper the magnitude of the mixing was estimated by calculating the smoothed variance of the northward component of velocity in the top 300 m of the ocean at 6°N.

Averages over the top few hundred metres are not available from the satellite data discussed here but, a closely related value, the geostrophic component of the surface velocity field can be calculated from the sea surface height. If $v$ is the northward component of geostrophic current at the ocean surface, then,

$$v = (g/f)\,\partial h/\partial x, \tag{1}$$

where $g$ is gravity, $f$ the Coriolis term, $h$ the sea surface height and $x$ the eastwards co-ordinate.

Let $\bar{v}$, be the smoothed value, produced by averaging over a range of longitudes and let $v_{rms}$, be the smoothed r.m.s. variance defined in a similar way. Then,

$$\begin{aligned} \bar{v} &= H(v), \\ v_{rms} &= H(|(v - \bar{v})|). \end{aligned} \tag{2}$$

where $H()$ is a smoothing filter. For the results presented here $H$ is a Hann filter with a width of 20°of longitude.

The results at 6°N, calculated using the above scheme, for both the satellite observations and the model are shown in Fig. 6.

In 1995 both figures show that the r.m.s velocity variances have maxima in the eastern Pacific during the second half of the year. Peak values are above $35\,\mathrm{cm\,s}^{-1}$ for both the model and observations but the model has a marginally greater range of longitudes with values above $15\,\mathrm{cm\,s}^{-1}$.

The results for 1996 also show increases during the second half of the year but in both cases the amplitudes are about $10\,\mathrm{cm\,s}^{-1}$ lower and the variance is more evenly spread across the ocean.

Late 1999 then shows an opposite extreme, with the values in the western Pacific based on observations reaching peak values of over $60\,\mathrm{cm\,s}^{-1}$ and with $45\,\mathrm{cm\,s}^{-1}$ over significant ranges of both space and time. The model shows a similar maximum during this period but peak values are nearer $50\,\mathrm{cm\,s}^{-1}$ and average values nearer $35\,\mathrm{cm\,s}^{-1}$.

However for the purposes of this paper the key period is during 1997 when the El Niño was developing. Figure 6 shows that as this was occurring, the model variance dropped to below $3\,\mathrm{cm\,s}^{-1}$ at times and was below $7\,\mathrm{cm\,s}^{-1}$ for long periods and over a large range of longitudes. In Webb (2018)

the corresponding velocity variance plot was taken to indicate reduced mixing, resulting in enhanced transport of the warmest surface water by the NECC.

However using the observed sea surface height, the r.m.s. variance is reduced during this period but it remains much larger than when calculated from the model data. Sometimes it drops below $7\,\mathrm{cm\,s}^{-1}$, but usually it is above $10\,\mathrm{cm\,s}^{-1}$ with peaks above $20\,\mathrm{cm\,s}^{-1}$. This discrepancy is discussed below.

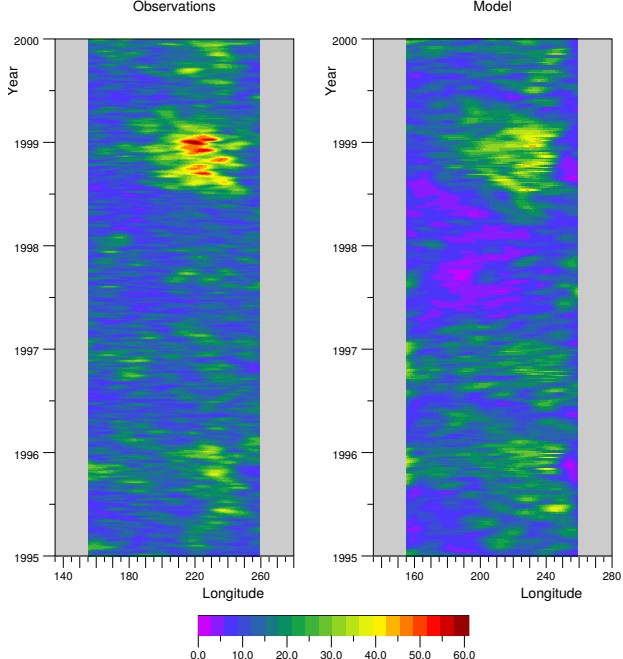

**Figure 6.** R.M.S variance of the northward component of ocean surface geostrophic velocity at 6°N. The values were calculated from the satellite altimeter and model sea surface height fields using Eqns. 1 and 2. Units are $\mathrm{cm\,s}^{-1}$.

## 6 Discussion

In a study of the 1982-1983 and 1997-98 El Niños, using data archived from a high resolution global ocean model, Webb (2018) proposed a number of mechanisms which helped trigger both events. The first was the increased strength of the NECC due to a larger sea level difference across the current than is normal. The second was an increase in the temperature of the water advected by the NECC due to reduced mixing of cooler water by tropical instability waves.

Prior to the original paper the realism of the model results had been checked by comparing the model prediction of temperatures in standard Niño regions with observations. However this study (Webb, 2016) did not investigate the realism

of the model SSH values and studied only part of the development in the surface temperature field in space and time.

The present paper has therefore concentrated on aspects of the temperature and SSH field that have most effect on the proposed mechanisms. For the temperature field, the results show that in the equatorial band, between 1995 and 2000, there is good agreement between the model and observations. In the autumn of 1997 during the main development phase of the El Niño, there is also good spatial agreement between the model and observations.

The comparison of SSH values at 6°N and at the Equator show similar good agreement. Together with the surface temperature comparison this does not prove the hypothesis about the role of the North Equatorial Trough, but it gives no reason to think it is wrong.

In the study of r.m.s. variance, the agreement during the development phase of the El Niño is not so good. During this period, the estimate based on the observed SSH data shows much more variance than that based on the model SSH. This implies that that tropical instability waves were more active in reality than in the model.

However this result is in conflict with the SST observations (Figs. 1 to 3) where temperature fluctuations due to tropical instability waves all but disappeared in the central Pacific. It is thus possible that there is some tropical instability wave activity during a strong El Niño but this is too weak to have a significant effect on the transport of warm water by the NECC.

Alternatively, given the potential errors in the altimeter data referred to earlier, it is possible that the differences in sea level variance during the growth period of the 1997-98 El Niño, results from some other cause such as the noise level of the gridded data.

*Code and data availability.* At the time of publication the model data is freely available at "http://gws-access.ceda.ac.uk/public/nemo/runs/ ORCA0083-N06/means/". The Nemo ocean model code and its documentation are available from "http://forge.ipsl.jussieu.fr/nemo/wiki/Users". The satellite temperature data is available from NASA/JPL PODAAC (https://podaac.jpl.nasa.gov/dataset/REYNOLDS_NCEP_L4_SST_OPT_INTERP_WEEKLY_V2). The satellite altimeter data is available from the Copernicus Marine Environment Monitoring Service (ftp://my.cmems-du.eu/Core/SEALEVEL_GLO_PHY_L4_REP_OBSERVATIONS_008_047/dataset-duacs-rep-global-merged-allsat-phy-l4/).

*Competing interests.* David Webb is on the advisory board of Ocean Science.

*Author contributions.* Dr Coward has been involved in developing the NEMO ocean model for many years and was responsible for setting up and running the 1/12°global ocean model and in curating the results. He also provided advice and help in analysing the data. Dr Snaith advised on the selection of satellite data sets, obtained access to the data and is primarily responsible for the discussion of the strengths and weaknesses of the individual datasets. The main author is responsible for the remainder of the paper.

*Acknowledgements.* I would like to thank the reviewers for their input which resulted in some important changes. This work contributed to and was aided by the research programme of the Marine Systems Modelling group at the UK National Oceanography Centre, part of the Natural Environment Research Council. The Natural Environment Research Council helped fund the investigation through the ODYSEA project (NE/M006107/1) and National Capability funding to NOC. Part of the analysis was carried out using the JASMIN Service at the UK Centre for Environmental Data Analysis, also funded by NERC.

Dr. W. Wimmer provided advice on the satellite datasets. The study depended heavily on the SST data provided by the NASA EOSDIS Physical Oceanography Distributed Active Archive Center at the Jet Propulsion Laboratory, Pasadena, CA (http://dx.doi.org/10.5067/GHGMR-4FJ01) and the SSH data provided by the Copernicus Marine Environment Monitoring Service (https://marine.copernicus.eu).

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
