# Peer review of "A comparison of Ocean Model Data and Satellite Observations of features affecting the growth of the NECC during the strong 1997-98 El Niño"

_Ocean Science, 2019_

## Referee Comment (RC1) · Anonymous Referee #1 · 17 Sep 2019

This study compares the simulation of the tropical Pacific by a high-resolution ocean model with available observations during the development of the extreme El Niño event of 1997-98. The motivation for this comparison stems from a previous study by the leading Author, who used the same ocean model to demonstrate the importance of the changes in the strength and temperature of the North Equatorial Counter Current (NECC) in the El Niño development.

I am not familiar with the previous paper by the Author, so that the mechanism by which the changes in the NECC at 5°N can produce large changes in the equatorial Pacific, where the largest SST and thermocline depth anomalies are ultimately found during

an El Niño, is not obvious from what presented in this paper. Although the mechanism may have been explained in depth in the lead Author's previous papers, to make this paper self-contained, more convincing arguments should be provided in this paper too, since the influence of the NECC on El Niño is the main motivation of this paper and guides all the diagnostics that are carried out by the Authors in this paper.

The mechanism for the development of an El Niño proposed by the Authors is an intriguing one, but in the strongly coupled series of events underlying the ENSO phenomenon, it is unclear how to identify causes and effects. During an ENSO event, the tropical Pacific undergoes a profound adjustment process, involving wind anomalies and wave propagation. In particular, anomalous warming in the equatorial Pacific causes a southward shift of the ITCZ with likely changes in the wind stress curl at the latitude of the mean ITCZ position. Changes in the NECC can be expected as a result of this adjustment, but it is unclear whether the NECC changes are actually the drivers of the El Niño development.

The Authors mention several time the annual Rossby wave. What causes this wave, why do they think that it is the main driver of the NECC changes, and why was the wave particularly strong in 1997? As mentioned before, Rossby waves in the tropical Pacific are the agent that allows the tropical ocean to adjust. Why so much emphasis on the annual Rossby wave?

The comparison between model and observations is very qualitative, except for the estimate of mixing in section 5. Much more could be done, including: 1) Support the interpretation of propagating anomalies as Rossby or Kelvin waves with an estimate of their phase speed; 2) Examine whether the changes in the NECC velocity, as estimated from the model, are consistent with the meridional gradient in sea level; 3) Compare SST and SSH hovmoeller diagrams to show that the warming seen along the equator is concurrent with equatorial Kelvin wave propagation, and 4) Estimate whether the changes in the strength of the NECC are indeed large enough to make an impact at 5N and along the equator.

Itemized comment, including typos:

1. Introduction, Lines 44-47. Why the deepening of the thermocline cannot produce surface warming? This should be briefly explained. 2. Introduction, Line 61. "where the" is repeated twice. 3. Section 1.1, line 3. How did the Authors assess that it was the "annual" Rossby wave to produce changes in the NECC? 4. Section 1.1, lines 25-30. The high sea level in the equatorial central Pacific discussed by Kug et al. (2009) only occurs during Central Pacific El Niño events. During Eastern Pacific events, like the extreme event consider here, the equatorial thermocline exhibits a very strong zonal dipole with deeper thermocline in the eastern Pacific and shallower thermocline in the western Pacific. 5. Section 1.2, Lines 53-55. SSH is important for its dynamical meaning, as it can be viewed as a proxy for thermocline depth and upper-ocean heat content. 6. Section 3, Line 26. Why was the annual Rossby wave unusually strong that year? 7. Section 3, P. 4, lines 12-16. In what way the chaotic nature of the waves is emphasized in difference plots? 8. Section 3, P. 3, lines 22-27. It is important to note that the two different stages of development of the 1997-98 El Nino have been related to different phases of Westerly Wind Burst (WWB) activity by several Authors (McPhaden 1999; Menkes et al. 2014; Capotondi et al. 2018, among others). How do the Authors reconcile the view they present in this paper with those previous studies? 9. Section 3.1, lines 42-44. The NECC can affect the ITCZ, but how is the perturbed ITCZ going to influence the warming in the eastern equatorial Pacific? 10. Section 4, line 43. How the "annual signal" was identified needs to be explained. 11. P. 6, lines 3-4. The increasing sea level in the west is typical of a developing La Niña, as it happened in 1998. 12. P. 7, line 40. I don't think that we are looking here at a model prediction, but at a model simulation. 13. P. 8, line 21. "that" is repeated twice.

References

Capotondi, A., P.D. Sardeshmukh, and L. Ricciardulli (2018), The nature of the stochastic wind forcing of ENSO, J. Climate, 31, 8081-8099.

McPhaden, M.J. (1999), Genesis and evolution of the 1997-98 El Niño, Science, 283, 950-954.

Menkes, C. E., M. Lengaigne, J. Vialard, M. Puy, P. Marchesiello, S. Cravatte, and G. Cambon, 2014: About the role of Westerly Wind Events in the possible development of an El Niño in 2014. Geophys. Res. Lett., 41, 6476-6483, doi:10.1002/2014GL061186.

---

## Author Comment (AC1) · 18 Sep 2019

I would like to thank the reviewer for taking the trouble with this paper and for his helpful comments, including those on the typos which always cause me problems.

Unfortunately I am about to be away for three weeks so I do not have time to give a detailed response soon but I would like to respond to a few points here, comments which might be of interest to other reviewers.

The reviewer states that he is not familiar with my previous paper on the 1982-83 and 1997-98 El Ninõs, but I would encourage the reviewer to read it for, although he or she

may strongly disagree with some or many of the conclusions, the paper answers many of the questions posed in the review.

For example the reviewer questions the emphasis on the annual Rossby wave, but the model results indicate that this is a significant feature of the tropical Pacific and that the resulting strengthening of the NECC, at the start of the strong El Ninõs, is sufficient to carry warm water from the western Pacific into the eastern Pacific. The results also indicate that this warm water is sufficient to trigger deep atmospheric convection at the latitude of the ITCZ.

The model results discussed in the earlier paper support the interpretation of the signal at 5N as an annual Rossby wave. There is also a section on the Kelvin waves which show that they have little or no effect on ocean surface temperatures - especially the temperatures need to trigger deep atmospheric convection. The equatorial Kelvin waves are there but they only thicken the surface layer. They may increase the upper ocean heat content but they do not result in any significant increase in SST.

The main weakness of the original paper is that it is based only on model results - and models can be (often are?) wrong. For this reason I thought it important to check the model further against observations - the result being this paper.

I could go all through the original arguments again but this would greatly lengthen the present paper and, given the potential for controversy, would if done properly move the paper away from its main focus - which is the agreement between the model and the observations.

This includes for example the development of high SST in the central Pacific during the strong 1997-98 El Ninõ. Despite what the reviewer states, this was also seen in the model results during the strong 1982-83 El Ninõ.

Anyway - many thanks for your interest, I'll provide a more detailed response later.

David Webb.

---

## Referee Comment (RC2) · Anonymous Referee #2 · 26 Oct 2019

This short paper presents a brief comparison between a model and data (sea level and SST) focusing on equatorial waves and tropical instability waves (TIW) during the 1997/98 El Niño event. The paper is a follow on of a previous recent paper by the lead author that also analyzes the same model and the same event. While I acknowledge the interest of investigating off-equatorial variability for understanding the build-up of heat content and the discharge process during strong El Niño events, it is not clear to me what is the specific motivations and objectives of the paper. The diagnostics are rather rudimentary and do not convey a clear message. The authors seem also to ignore the existing literature on this event that has been extensively documented and investigated. They should clarify what is their specific contribution compared to

previous studies and resolve some methodological issues (see specific comments).

Specific comments:

Abstract "The results provide additional confidence in the oceanic mechanisms which model analysis implicated as being responsible for the development of both the 1982-83 and the 1997-98 El Niño". This is quite a vague statement. The abstract should provide some hints of what are the results.

Introduction It does not convey a clear motivation and there is no references to relevant works (McPhaden (1999), Boulanger and Menkes, (1999), Vialard et al. (2001) amongst many others, see also all the literature on the ENSO-TIW interaction (see introduction of Holmes et al. (2019) for instance)

Holmes R. M., S. McGregor, A. Santoso and M.H. England (2019) Contribution of Tropical Instability Waves to ENSO Irregularity, Climate Dynamics, 52, 1837-1855.

Boulanger, J.-P., and C. Menkes, Long equatorial wave reflection in the Pacific Ocean during the 1992-1998 TOPEX/POSEIDON period, Clim. Dyn. 15, 205-225, 1999.

McPhaden, M. J., Genesis and evolution of the 1997-1998 El Niño, Science, 283, 950-954, 1999.

Vialard, J., C. Menkes, J.-P. Boulanger, P. Delecluse, E. Guilyardi , M. J. McPhaden et G. Madec, Oceanic mechanisms driving the SST during the 1997-1998 El Niño, J. Phys. Oceanogr., 31, 1649-1675, 2001.

The statement "The study concentrated on the strong El Niños of 1982-83 and 1997-97 and found that equatorial Kelvin waves had no significant effect on the surface temperature of the eastern Pacific." is surprising. It is recognized that the Kelvin wave during El Niño produce vertical advection of anomalous temperature, a process refers as the thermocline feeback and shown to be dominant in the eastern equatorial Pacific in previous ENSO studies. Model data comparison The model has no assimilation of data so it is difficult to compare model and observations in terms of TIW, the model simulatOSD
ing eddies that are not necessarily collocated with observations owing to their chaotic nature. So the comparison should be based on statistics rather than the visual inspection of Hovmöller diagrams (e.g. Figure 2). See for instance An (2008) for relevant diagnostics for TIW activity.

An, S.-I., 2008 : Interannual Variations of the Tropical Ocean Instability Wave and ENSO, J. Climate. 21, 3680-3686.

Figure 4: It is not really possible to see an equatorial Kelvin wave at  $6^{\circ}$ N; its amplitude would be very weak. Also the difference between model and observation is not relevant here unless you focus on the low frequencies (periods >  $\sim$ 60 days), which would require filtering the data. Comparison should be done on anomalies relative to the mean climatology, otherwise this is just emphasizing the differences in seasonal cycle. If the authors want to discuss Kelvin and Rossby wave contribution to sea level anomalies, I suggest that they project sea level on the theoretical equatorial wave structures (see Boulanger and Menkes (1995) for the method).

Boulanger, J.-P., et C. Menkes, Propagation and reflection of long equatorial waves in the Pacific ocean during the 1992-1993 El Niño, J. Geophys. Res., 100, 25041-25059, 1995.

OSD

---

## Author Comment (AC2) · 28 Oct 2019

Again many thanks for your review. In the following, the reviewer's comments are in **bold text** and my response in normal text.

Although the mechanism may have been explained in depth in the lead author's previous papers, to make this paper self-contained, more convincing arguments should be provided in this paper too.

I accept the referee's point and will try to, briefly, explain the importance of high sea surface temperatures and the roles of the annual Rossby wave, tropical instability waves (eddies?) and sea level near the Equator in the mechanisms discussed in the original paper.

The mechanism for the development of an El Niño proposed by the authors is an intriguing one, but in the strongly coupled series of events underlying the ENSO phenomenon, it is unclear how to identify causes and effects. During an ENSO event, the tropical Pacific undergoes a profound adjustment process, involving wind anomalies and wave propagation. In particular, anomalous warming in the equatorial Pacific causes a southward shift of the ITCZ with likely changes in the wind stress curl at the latitude of the mean ITCZ position. Changes in the NECC can be expected as a result of this adjustment, but it is unclear whether the NECC changes are actually the drivers of the El Niño development.

This is really a comment on the original paper. From what I can tell, since Wyrtki's time people have been trying to separate causes and effects in the El Niño system without a lot of success. The previous paper had the advantage of access to the results from an ocean model which has much higher resolution than normal in both the horizontal and vertical directions. Thus it is possible that it would emphasise connections which were hidden by resolution problems in earlier studies.

However it might have also introduced new errors - and that is the reason for the present paper - to check key oceanic features which were central to the arguments in the previous paper. I think the previous paper gives reasonable causal arguments that changes in the the annual Rossby wave in the western Pacific and changes in the strength of the tropical instability waves are enough to allow the NECC to transport warm water (>28 °C) from the western to eastern Pacific at latitudes near the ITCZ.
Such water is known to be warm enough to trigger deep atmospheric convection (see the paper by Evans and Webster).

It is possible that SST values near 26  $^{\circ}$ C on the Equator are more important than 28  $^{\circ}$ C near the ITCZ but that is for someone else to check.

The authors mention several times the annual Rossby wave. What causes this wave, why do they think that it is the main driver of the NECC changes, and why was the wave particularly strong in 1997? As mentioned before, Rossby waves in the tropical Pacific are the agent that allows the tropical ocean to adjust. Why so much emphasis on the annual Rossby wave?

The emphasis on the annual Rossby wave arises because in both 1982 and 1997 it propagated further than normal into the western Pacific. This resulted in a greater sea level difference across the latitudes of the NECC, which because of geostrophy increased the transport of the NECC. This meant that it carried eastwards a greater amount of warm pool water than normal - starting the process which eventually resulted in water with temperatures greater than 28 °C reaching the eastern Pacific.

The wave is thought be be generated in the eastern Pacific, partly by winds blowing across the Isthmus of Panama. As it propagates across the Pacific it my be modified by local winds. However I have not been able to find any modern authoritative paper or review of the wave, its generation and propagation.

The comparison between model and observations is very qualitative, except for the estimate of mixing in section 5.

OSD
Yes, this is a weakness of the paper. However the eye is a pretty good data processor and the figures have been chosen and constructed such that if there are significant differences they should show up - as they do in the plots which aim to measure the strength of the topical instability waves. An analysis which gives a single number can look fine but hide a wealth of errors.

The review makes a number of suggestions :

Much more could be done, including: 1) Support the interpretation of propagating anomalies as Rossby or Kelvin waves with an estimate of their phase speed; 2) Examine whether the changes in the NECC velocity, as estimated from the model, are consistent with the meridional gradient in sea level; 3) Compare SST and SSH Hovmoeller diagrams to show that the warming seen along the equator is concurrent with equatorial Kelvin wave propagation, and 4) Estimate whether the changes in the strength of the NECC are indeed large enough to make an impact at 5N and along the equator.

These are really comments on the physics discussed in the original paper (which was long enough) - whereas the present paper is focusing on the equally important problem of model accuracy. If the comparisons with observations show that the model is wrong in describing key features then the arguments of the previous paper are irrelevant anyway.

For (1) I did some rough comparisons and was confident that these were Rossby and Kelvin waves but further analysis is always possible. For (2) a lot of effort had been spent improving the NEMO physical model. The model is also widely used so I believe that if there was any problem with geostrophy in the model it would have shown up

OSD
elsewhere by now. (3) I do not think the warming seen along the Equator is consistent with equatorial Kelvin wave propagation. This can be seen by comparing figure 6 and 19 of the original paper and figure 1 and 5 of the present paper. The Kelvin waves are the fast waves seen in figure 5 of the present paper. (4) The particle tracking plots of the original paper showed that the NECC was able to transport warm pool water into the eastern Pacific. Some of this will have displaced water towards the Galapagos but the main reason for warming on the Equator there is probably a result of reduced upwelling due to a reduction in wind generated Ekman divergence.

Reviewer's Detailed comments:

**1. Why the deepening of the thermocline cannot produce surface warming? This should be briefly explained.**

Not really a question for the present paper. At a fixed depth within the thermocline, deepening of the thermocline will produce warming due to the descending warmer water. However at the surface there is nothing to descend. The only way the surface layer can warm is through horizontal advection or by local processes, such as increased heat flux into the ocean. The Webb (2016) paper indicated that in the Nino regions studied, the temperature gain during the development of the El Niño was not due to local processes.

3. How did the Authors assess that it was the "annual" Rossby wave to produce changes in the NECC?

Not really for the present paper, but see the SSH figures.

OSD
4. The high sea level in the equatorial central Pacific discussed by Kug et al. (2009) only occurs during Central Pacific El Niño events.

The observational evidence (Fig. 5) shows that this is not always true.

4.1 During Eastern Pacific events, like the extreme event consider here, the equatorial thermocline exhibits a very strong zonal dipole with deeper thermocline in the eastern Pacific and shallower thermocline in the western Pacific.

If we assume that sea level is a measure of thermocline depth, then Fig 5 shows that there is a deeper thermocline in the eastern Pacific and shallower thermocline in the western Pacific - however that is not the full story.

5. SSH is important for its dynamical meaning, as it can be viewed as a proxy for thermocline depth and upper-ocean heat content.

I agree.

6. Why was the annual Rossby wave unusually strong that year?

I do not know why it was strong that year.

7. In what way the chaotic nature of the waves is emphasized in difference plots?
Let the variances in the two plots, measured relative to the same time and place, be V1 and V2. If the processes are random then the variance of their difference is expected to be V1+V2, and the r.m.s. value is the square root of V1+V2. This is larger than the figures for the individual plots.

8. It is important to note that the two different stages of development of the 1997-98 El Niño have been related to different phases of Westerly Wind Burst (WWB) activity by several Authors (McPhaden 1999; Menkes et al. 2014; Capotondi et al. 2018, among others). How do the Authors reconcile the view they present in this paper with those previous studies?

Not really for the present paper. However Fig. 32 of the original paper is a plot of the easterly wind stress along the Equator for the period 1995 to 2000. The data comes from the ECMWF reanalysis and during 1997 it shows a number of westerly wind bursts.

Figure 5 of the present paper shows the satellite observations of sea level along the equator and also the model response. Both show the Kelvin waves generated by the westerly wind bursts.

Figure 1 of the present paper shows the satellite observations of sea surface temperature along the equator and also the model response. Both plots show that the temperature anomalies propagate much more slowly than the equatorial Kelvin waves. They also show little or no evidence of Kelvin wave activity.

9. The NECC can affect the ITCZ, but how is the perturbed ITCZ going to influence the warming in the eastern equatorial Pacific?
Not really for the present paper. However I expect it is through a reduction in the winds along the equator and the resulting reduction in wind driven upwelling along the Equator (see earlier comment).

**10. How the "annual signal" was identified needs to be explained.**

Not really for the present paper but see the plots. Some of the plots were made for the whole period 1957-2009. Something that usually occurred at the same time every year during this period was assumed to be an annual signal.

**11. The increasing sea level in the west is typical of a developing La Niña, as it happened in 1998.**

In 1984 the model behavior is similar.

12. p.7 line 40. I don't think that we are looking here at a model prediction, but at a model simulation.

I agree.

In conclusion thanks for the comments. Most of them however are related to the content of the previous paper - which I accept remains an area of debate and which I am happy to continue elsewhere. Therefore, and unless the editor advises me differently, in the revised paper I shall concentrate on a better summary of the earlier results and a better explanation of the reasons for the present paper. OSD
Regards,

David Webb.
OSD

---

## Author Comment (AC3) · 31 Oct 2019

I would like to thank the reviewer for taking on this paper and for the comments and references.

As I understand it the comments are of two types. The first concerns just the present paper and mainly concerns how difficult it is for the reader to understand the background and reasons for the paper's publication. The second is a criticism, valid for both papers, of a general lack of external references.

I should say now that the lack of references results from a decision made when I started this study. As the reader might know, in the past I have been involved in a number of studies of the Pacific Ocean, mainly modeling but also experimental, including jointly leading a study of the far western equatorial Pacific. None of this has actively involved the El Niño but of course I have been very aware of El Niño research and ideas.

For the present project I decided to get involved in studying a model run that my exgroup was involved with and to use it to understand more about the El Niño. I am of course aware that the process has been studied at great expense for over 50 years and has produced hundreds if not thousands of papers. But, at the same time I am concerned by the lack of a comparable improvement in physical understanding during this period and the fact that serious researchers can still describe the El Niño, in a new-age way, as a subtle interaction of ocean and atmosphere, with little hard physics and little embarrassment.

I therefore tried to make progress by being different. I did this by focusing on the one model ocean and by trying to make sense of the physics without the constraints of any particular theoretical model or line of research. If it had failed you would have heard no more - but it had some success, one result of which is that in Webb (2018) and this paper there are few references - except ones that help better explain the model's behavior.

Now you may be unhappy about this but it is done to keep the papers focused. There will be time later to discuss how and why other approaches agree or disagree.

In the following, the reviewer's comments are in **bold text** and my response in normal text.

**OSD**
This short paper presents a brief comparison between a model and data (sea level and SST) focusing on equatorial waves and tropical instability waves (TIW) during the 1997/98 El Niño event. The paper is a follow on of a previous recent paper by the lead author that also analyzes the same model and the same event. While I acknowledge the interest of investigating off-equatorial variability for understanding the build-up of heat content and the discharge process during strong El Niño events, it is not clear to me what is the specific motivations and objectives of the paper.

As with the first reviewer I accept that the introduction needs to be made clearer.

**The diagnostics are rather rudimentary and do not convey a clear message.**

I am surprised by this comment. The eye can be a good data processor and in this paper the figures usually show remarkably good agreement between the model and observations. See for example the equatorial Kelvin waves triggered by westerly wind bursts during the development of an El Niño.

The authors seem also to ignore the existing literature on this event that has been extensively documented and investigated.

Point discussed above.

They should clarify what is their specific contribution compared to previous studies and resolve some methodological issues (see specific comments).

**OSD**
The aim of this paper is to test key areas of the model's response against observations. This needs to be made clearer in the revised paper. The comparison has been restricted by the limited amount of satellite data available for the 1997-1998 period.

The paper does not try and prove that the mechanisms discussed in Webb (2018) are correct, rather it is a test to see if there are major faults in the model which means that their justification is wrong. Thus if anything it is trying to disprove the previous results and conclusions. The fact that no major problems are found in the key areas studied gives some additional confidence that the Webb (2018) conclusions are not based on a poor version of the numerical model.

Specific comments:

Abstract "The results provide additional confidence in the oceanic mechanisms which model analysis implicated as being responsible for the development of both the 1982- 83 and the 1997-98 El Niño". This is quite a vague statement. The abstract should provide some hints of what are the results.

This will to be improved.

Introduction It does not convey a clear motivation and there is no references to relevant works (McPhaden (1999), Boulanger and Menkes, (1999), Vialard et al. (2001) amongst many others, see also all the literature on the ENSO-TIW interaction (see introduction of Holmes et al. (2019) for instance).

The motivation needs to be made clearer and I will try and do this. The references
included in the review are of interest in any discussion of the ocean's role in the El Niño and for that reason I include a few comments below. However the only role I see for such references in a model-data comparison paper is to emphasise the differences in the conclusions reached and to emphasise the importance of checking that the differences are not due to errors in the present model.

**Holmes R. M., S. McGregor, A. Santoso and M.H. England (2019) Contribution of Tropical Instability Waves to ENSO Irregularity, Climate Dynamics, 52, 1837-1855.**

Interesting, but as shown in Fig.6 of the present paper and Fig. 14 of Webb (2018), TIW amplitudes are generally very low during the year in which strong El Ninõs develop.

**Boulanger, J.-P., and C. Menkes, Long equatorial wave reflection in the Pacific Ocean during the 1992-1998 TOPEX/POSEIDON period, Clim. Dyn. 15, 205-225, 1999.**

Thank you for bringing this paper to my attention. As I discuss below I am not completely happy with a wave guide expansion. Maybe the strong coupling of the first and third Rossby wave modes reflects the weakness of this approach.

McPhaden, M. J., Genesis and evolution of the 1997-1998 El Niño, Science, 283, 950-954, 1999.

An 'authorative' paper which now appears to contain many misleading statements. To quote from just the second paragraph:

**OSD**
A weakening and reversal of the trade winds in the western and central equatorial Pacific led to the rapid development of unusually warm sea-surface temperatures (SSTs) east of the international date line in early 1997 (Figs. 1 and 2).

The phase 'led to' implies causes. The work reported in Webb (2018) concluded that both changes were a result of increased advection of warm water by the NECC.

The western Pacific warm pool (surface waters greater than about 29  $^{\circ}$ C) migrated eastward with the collapse of the trade winds,

This will be taken by some to imply that the collapse of the trade winds was the cause of the migration. Webb(2018) finds that it was the other way around - the migration of the warm pool, due to the NECC, led and caused the collapse of the trade winds in the western Pacific.

and the equatorial cold tongue — the strip of cool water indicative of equatorial upwelling that normally occupies the eastern and central Pacific between the coast of South America to the international date line — failed to develop in boreal summer and fall 1997.

This I would agree with if it had also pointed out the reduced winds in the central and western Pacific (McPhaden, Fig. 1a) and that this reduction would have resulted in reduced upwelling.

Vialard, J., C. Menkes, J.-P. Boulanger, P. Delecluse, E. Guilyardi, M. J. McPhaden and G. Madec, Oceanic mechanisms driving the SST during the 1997-1998 El
**Niño, J. Phys. Oceanogr., 31, 1649-1675, 2001.**

I like this paper especially the care taken to validate the model before use. The model is similar to that used in Webb (2018) but uses lower resolution both horizontally and vertically. Despite this, many of the results appear similar, an example being the reduction in TIW strength during the development of an El Niño.

Where I would disagree is in the interpretation of the results. As an example, the paper says that "some of the ocean processes in early 1997 are associated with a strong Madden-Julian Oscillation", but similar events are seen in the model results for 1982-83 when the oscillation was very weak. The importance of meridional advection is mentioned but not the currents involved. The interpretation focuses on equatorial Kelvin waves and when, for example, discussing the deeper thermocline in the east in early 1997 it gives the same weight to the downwelling Kelvin waves as it gives to the "weaker winds" in the east.

Kelvin waves, like other waves, cannot generate a net transport sufficient to give a step change in the thermocline depth in the eastern Pacific - the Stoke's drift due to the waves being insufficient. In contrast weaker trade winds in the eastern equatorial Pacific will result in reduced upwelling there and so allow a new thermal balance to develop and/or allow the advective inflow of warmer near-surface waters from the surrounding ocean.

The statement "The study concentrated on the strong El Niños of 1982-83 and 1997-97 and found that equatorial Kelvin waves had no significant effect on the surface temperature of the eastern Pacific." is surprising. It is recognized that the Kelvin wave during El Niño produce vertical advection of anomalous temper-

**OSD**
**ature, a process refers as the thermocline feedback and shown to be dominant in the eastern equatorial Pacific in previous ENSO studies.**

This statement relates to the conclusions of Webb (2018) and should not concern the present comparison of model results with observations. The point is discussed above. In addition Fig. 5 shows equatorial Kelvin waves, triggered by westerly wind bursts, which arrive in the eastern Pacific in the second quarter of 1997. These are the same wind bursts as discussed by Villard et al (2001) and discussed above. Figure 1 of the present paper shows a slight warming in the same region before and at the start of this period. If, despite what was said above, I am wrong and the two processes are connected then I would argue that the warming is not significant in the sense that the temperatures are not high enough to trigger deep atmospheric convection.

These figures and the wind stress figure in the original paper show that the Kelvin waves are generated by westerly wind bursts within the warm region of ocean but they appear to be unconnected to the rate at which the warm region progresses across the ocean.

Model data comparison: The model has no assimilation of data so it is difficult to compare model and observations in terms of TIW, the model simulating eddies that are not necessarily collocated with observations owing to their chaotic nature. So the comparison should be based on statistics rather than the visual inspection of Hovmöller diagrams (e.g. Figure 2). See for instance An (2008) for relevant diagnostics for TIW activity.

I'll quantify the variance in some of the key regions and add the values to the text. On the point of data assimilation I would have thought that if data was assimilated then

**OSD**
usually (a) any comparison between model and observations would be suspect and (b) any inferences concerning cause and effect would be suspect.

**Figure 4: It is not really possible to see an equatorial Kelvin wave at 6°N; its amplitude would be very weak.**

In the study of the 1982-1983 El Niño, the model showed a large number of significant changes which occur all across the ocean during a very short period at the end of 1982. Similar changes occur at the end of 1997. These occur not only at the Equator (Webb (2018) Figs. 7, 15, 19, 28, 31) but also at  $6^{\circ}N$  (Webb (2018), Figs. 7, 11, 20, 21, 28, 29, 31). Because of the speed at which the change occurs I have concluded that this is the effect of a Kelvin wave.

The latitudinal extent of a Kelvin wave does depend on stratification, but in the example given by Boulanger and Menkes (1995), which you refer to, the Kelvin wave does affect 6 °N.

In the revised paper I will add a note referring to the original paper, saying that the results imply, but don't prove, that the feature is due to a Kelvin wave.

Also the difference between model and observation is not relevant here unless you focus on the low frequencies (periods  $>\sim$ 60 days), which would require filtering the data.

I do not understand the filtering argument but I agree that the figure does not add a lot. However as Figs. 1 and 2 include differences there seemed no reason not to include
it here. One advantage of including it is that then all three figures will be printed at a similar size in the final paper.

Comparison should be done on anomalies relative to the mean climatology, otherwise this is just emphasizing the differences in seasonal cycle. If the authors want to discuss Kelvin and Rossby wave contribution to sea level anomalies, I suggest that they project sea level on the theoretical equatorial wave structures (see Boulanger and Menkes (1995) for the method).

I disagree with the use of anomalies, unless nothing else is available, because they are often misleading. The web contains many pictures of El Niño temperature anomalies (and associated sea level anomalies) in the cold pool upwelling region, where maybe the temperature has increased from 19C to 26C. They look very dramatic and are sometime used to 'explain' why the atmospheric circulation changes. However the effect of the temperature change on deep atmospheric convection and the El Niño is probably insignificant compared with a temperature increase from 27C to 29C near the latitudes of the ITCZ.

In the case of Fig. 4 the plot of absolute values emphasizes the fact that the changes in western Pacific sea levels during an El Niño are coordinated with (not necessarily caused by) the timing of the annual Rossby wave. If an anomaly was used this would not be so clear.

On the question of using modes of the equatorial waveguide, waveguide modes higher than the base mode are often represented as pairs of simple waves which cross each other at a small angle at the center of the waveguide, in this case the Equator. They then get reflected or refracted at two critical boundaries (latitudes) decaying exponen-
tially outside the boundaries and combining to generate standing waves between the boundaries (as seen in the Rossby wave solutions).

For the ocean this implies that energy crosses the Equator at some point. I do not know of any examples of this. For very long east-west wavelengths this may be because the intersection angle is very shallow, the different modes having the same limiting phase and group velocities - so a description in terms of purely westward traveling features is equally valid. The fact that equatorial crossing is not seen at shorter east-west length scales is puzzling, unless stratification and currents near the equator are acting as barriers to the flux of energy.

In conclusion I agree that the introduction and abstract need improving and that a number of other points need attention. Thank you for your comments and the chance to highlight some of the differences between the conclusions of Webb (2018) and previous analyses of the 1997-1998 event.

Regards,

David Webb.

---

## Author Response (AR1)

Cover Note
========

As promised in my responses to the reviewers I have modified the abstract and introduction to give more information about the results from my earlier paper. I have also corrected the minor errors that they pointed out and a number of additional small errors that I found.

In addition I have made two additional sets of changes resulting in part from the reviewers' comments.

The first of these concerns the large changes that occur all across the ocean at the end of 1997 and the start of 1998. Similar changes are seen at the end of 1982 and the start of 1983. In the original text I had put this down as the result of a Kelvin wave, but on reflection I decided that I had made the same mistake that I had accused others of having - that is explaining step like changes as being the result of simple propagating waves.

Looking again at the model results I found that in the model forcing field this was a time when the NE trades reached and sometimes crossed the Equator. I also found a paper, referred to in the new text, which showed that the ITCZ crossed the equator the start of 1983 and 1998 and at no other time during the period studied. I have therefor revised the text to point this out.

This also ties in with another study, which I am currently involved in, which indicates that the apparent extension of the annual Rossby wave into the western Pacific may be just that - an apparent extension. I have therefor changed the text to indicate that this may be a local western Pacific event.

 Regards,

 David.

[revised manuscript text omitted]

---

## Author Response (AR2)

```
03_Response
==========
```

**1.  Response to Reviewers**

**1.1  Reviewer 1**

After the first set of reviews, the section on the mechanisms suggested by Wyrtki and Webb was expanded.  This inevitably affected the balance of the paper, resulting in the reviews comment that "the paper argues" etc.

To try and restore the balance, a paragraph has been added to the Abstract emphasing that the paper is a targeted study to see if the model results could be shown to be in serious error and thus give no support to the mechanisms.  I have also emphasised this in the new Additional Tests section at the end of the paper.

The reviewer's second point, that I agree with, is that the present comparison is mainly qualitative  (although some quantitative comparisons are included).  To cover this point the reviewer suggests that further tests be suggested.  I am happy with this and have added four suggestions in the Additional Tests section.

On the third point, the title has been changed as suggested.

**Reviewer 2**

The main point made here is the problem of Causality, whether the mechanisms/hypotheses suggested are causal or that the apparent relationships are due to other processes.

I cannot deal with this in the short period given for a minor revision but I have added two suggestions in the Additional Tests which should help to identify more precisely the causes and the effects.  In any case I do not think that there is enough information in the 1997-1998 satellite data to say anything about causality.

Regards,

David Webb.

**2.  Changes in the text**

**2.1  Title changed**

It is now "A comparison of Ocean Model Data and Satellite Observations of features affecting the growth of the NECC during the strong 1997-98 El  Ni\~no".

**2.2.  Addition to abstract**

```
* * *
```
The results of this paper should not be taken as providing proof of the hypotheses of \cite{Wyrtki_1973, Wyrtki_1974} or \cite{Webb_2018b} but instead as a failure of a targeted study, using satellite observations, to disprove the hypotheses.
```
* * *
```

**2.3.  Minor changes in text - see pdf differences below.**

**2.4.  Addition of 'Additional Tests' section at end of Discussion**

```
* * *
\subsection{Additional Tests}
```

In physics, hypotheses can only be disproven, but if well targeted test fail to do this then one might start having some confidence in the ideas.  In the present case the targets have been aspects of the model results which were critical for the development of the ideas of \cite{Webb_2018b}.  The present set of comparisons, despite using some of the best data sets available, have failed to show that the computer model results, and thus the hypotheses, are seriously in error.

However further tests are needed, to ensure that the model results are realistic and that mechanisms proposed are causal.  One check on reality would be to repeat the comparisons of this paper for the 2015-2016 El Ni\~no when much more satellite data is available.  Unfortunately the high resolution computer model run stopped at the end of 2015, but changes during the El Ni\~no growth phase can still be investigated.

A second is to compare the strength of the NECC and the tropical instability eddies in the model with temperature and current meter observations from the region.  The TOGA-TAU buoy array \citep{Hayes_etal_1991} covers the region of interest and has measured temperatures in the top few hundred metres of the ocean since 1992.  Unfortunately there are no comparable velocity measurements in the central Pacific, except at the Equator, but one might hope that this will change.

The question of causality may be tackled by analysing the model results in more detail.  Although \cite{Webb_2018b} was mainly concerned with qualitative comparisons, \cite{Webb_2016} included quantitative estimates of surface heating in the eastern Pacific and this could be extended to include the horizontal and vertical advection of heat into the surface layers at both the Equator and the latitude of the NECC.  This should give a more quantitative understanding of the role of waves and advection near to the Equator and the effect of tropical instability eddies on the volume transports and temperatures of the warmest water masses transported by the NECC.

Finally there is the question of where and why deep atmospheric convection occurs in the central and eastern Pacific.  The study by \cite{Evans&Webster_2014} is notable in the way it excludes much of the Pacific.  A similar study is required which corrects this and which investigates the pathways and preconditioning of the air masses by the ocean prior to the occurrence of deep convection events there.

This would help identify the key ocean regions responsible for the changes in deep atmospheric convection that occur at the start of a strong El Ni\~no.
* * *
**A  comparison of Ocean Model  Data and Satellite Observations  of features affecting the  growth of the NECC during the strong 1997-98 El Niño**

**David J. Webb, Andrew C. Coward, and Helen M. Snaith**

National Oceanography Centre, Southampton SO14 3ZH, U.K.

**Correspondence:** D.J.Webb (djw@noc.ac.uk)

**Abstract.**

Descriptions of the ocean's role in the El Niño usually focus on equatorial Kelvin waves and the ability of such waves to change the mean thermocline depth and the sea surface temperature (SST) in the central and  eastern Pacific.

In contrast, starting from a study of the transport of water with temperatures greater than 28°C, sufficient to trigger deep atmospheric convection, Webb (2018) found that during the strong El Niños of 1983-1984 and 1997-1998, advection by the North Equatorial Counter Current (NECC) had a much greater impact on sea surface temperatures than processes occurring near the Equator.

Webb's analysis, which supports the scheme proposed by Wyrtki (1973, 1974), made use of archived data from a high resolution ocean model. Previously the model had been checked in a preliminary comparison against SST observations in the equatorial Pacific, but given the contentious nature of the new analysis, the model's behaviour in key areas need to be checked further against observations.

In this paper this is done for the 1987-1988 El Niño, making use of satellite observations of SST and sea level. SST is used to check the movement of warm water near the Equator and at the latitudes of the NECC. Sea level is used to check the model results at the Equator and at 6°N in the North Equatorial Trough. Sea level differences between these latitudes affect the transport of the NECC, the increase transport at the start of each strong El Niño being associated with a drop in sea level at 6°N in the western Pacific. Later rises in sea level at the Equator increases the transport of the NECC in mid-ocean.

The variability of sea level at 6°N is also used to compare the strength of tropical instability waves (TIWs) in the model and in the observations. The model showed that in a normal year these act to dilute the temperature in the core of the NECC. However their strength declined during the development of the strong El Niños, allowing the NECC to carry warm water much further than normal across the Pacific.

The results of this paper should not be taken as providing proof of the hypotheses of Wyrtki (1973, 1974) or Webb (2018) but instead as a failure of a targeted study, using satellite observations, to disprove the hypotheses.

[revised manuscript text omitted]

**6.1 Additional Tests**

In physics, hypotheses can only be disproven, but if well targeted test fail to do this then one might start having some confidence in the ideas. In the present case the targets have been aspects of the model results which were critical for the development of the ideas of Webb (2018). The present set of comparisons, despite using some of the best data sets available, have failed to show that the computer model results, and thus the hypotheses, are seriously in error.

However further tests are needed, to ensure that the model results are realistic and that mechanisms proposed are causal. One check on reality would be to repeat the comparisons of this paper for the 2015-2016 El Niño when much more satellite data is available. Unfortunately the high resolution computer model run stopped at the end of 2015, but changes during the El Niño growth phase can still be investigated.

A second is to compare the strength of the NECC and the tropical instability eddies in the model with temperature and current meter observations from the region. The TOGA-TAU buoy array (Hayes et al., 1991) covers the region of interest and has measured temperatures in the top few hundred metres of the ocean since 1992. Unfortunately there are no comparable velocity measurements in the central Pacific, except at the Equator, but one might hope that this will change.

The question of causality may be tackled by analysing the model results in more detail. Although Webb (2018) was mainly concerned with qualitative comparisons, Webb (2016) included quantitative estimates of surface heating in the eastern Pacific and this could be extened to include the horizontal and vertical advection of heat into the surface layers at both the Equator and the latitude of the NECC. This should give a more quantitative understanding of the role of waves and advection near to the Equator and the effect of tropical instability eddies on the volume transports and temperatures of the warmest water masses transported by the NECC.

Finally there is the question of where and why deep atmospheric convection occurs in the central and eastern Pacific. The study by Evans and Webster (2014) is notable in the way it excludes much of the Pacific. A similar study is required which corrects this and which investigates the pathways and preconditioning of the air masses by the ocean prior to the occurrence of deep convection events there.

This would help identify the key ocean regions responsible for the changes in deep atmospheric convection that occur at the start of a strong El Niño.

*Code and data availability.* At the time of publication the model data is freely available at "http://gws-access.ceda.ac.uk/public/nemo/runs/ ORCA0083-N06/means/". The Nemo ocean model code and its documentation are available from "http://forge.ipsl.jussieu.fr/nemo/wiki/Users". The satellite temperature data is available from NASA/JPL PODAAC (https://podaac.jpl.nasa.gov/dataset/REYNOLDS_NCEP_L4_SST_OPT_INTERP_WEEKLY_V2). The satellite altimeter data is available from the Copernicus Marine Environment Monitoring Service (ftp://my.cmems-du.eu/Core/SEALEVEL_GLO_PHY_L4_REP_OBSERVATIONS_008_047/dataset-duacs-rep-global-merged-allsat-phy-l4/).

*Competing interests.* David Webb is on the advisory board of Ocean Science.

*Copyright statement.* The works published in this journal are distributed under the Creative Commons Attribution 4.0 License. This licence does not affect the Crown copyright work, which is re-usable under the Open Government Licence (OGL). The Creative Commons Attribution 4.0 License and the OGL are interoperable and do not conflict with, reduce or limit each other.

*Author contributions.* Dr Coward has been involved in developing the NEMO ocean model for many years and was responsible for setting up and running the 1/12° global ocean model and in curating the results. He also provided advice and help in analysing the data. Dr Snaith advised on the selection of satellite data sets, obtained access to the data and is primarily responsible for the discussion of the strengths and weaknesses of the individual datasets. The main author is responsible for the remainder of the paper.

*Acknowledgements.* I would like to thank the reviewers for their input which resulted in some important changes. This work contributed to and was aided by the research programme of the Marine Systems Modelling group at the UK National Oceanography Centre, part of the Natural Environment Research Council. The Natural Environment Research Council helped fund the investigation through the ODYSEA project (NE/M006107/1) and National Capability funding to NOC. Part of the analysis was carried out using the JASMIN Service at the UK Centre for Environmental Data Analysis, also funded by NERC.

Dr. W. Wimmer provided advice on the satellite datasets. The study depended heavily on the SST data provided by the NASA EOSDIS Physical Oceanography Distributed Active Archive Center at the Jet Propulsion Laboratory, Pasadena, CA (http://dx.doi.org/10.5067/GHGMR-4FJ01) and the SSH data provided by the Copernicus Marine Environment Monitoring Service (https://marine.copernicus.eu).